# Evolving the Olfactory System

**Guangyu Robert Yang**[*] (gyyang.neuro@gmail.com), **Peter Yiliu Wang**[*] (peterwang724@gmail.com),
**Yi Sun (yisun@math.columbia.edu), Ashok Litwin-Kumar (ak3625@columbia.edu),**
**Richard Axel (ra27@columbia.edu), L.F. Abbott (lfa2103@columbia.edu),**
Columbia University, New York, NY 10027 USA
* equal contributions

## Abstract

**Flies and mice are species separated by 600 million years of evolution, yet have evolved olfactory systems that share many similarities in their anatomic and functional organization. What functions do these shared anatomical and functional features serve, and are they optimal for odor sensing? In this study, we address the optimality of evolutionary design in olfactory circuits by studying artificial neural networks trained to sense odors. We found that artificial neural networks quantitatively recapitulate structures inherent in the olfactory system, including the formation of glomeruli onto a compression layer and sparse and random connectivity onto an expansion layer. Finally, we offer theoretical justifications for each result. Our work offers a framework to explain the evolutionary convergence of olfactory circuits, and gives insight and logic into the anatomic and functional structure of the olfactory system.**

## Introduction

Over the last two decades, both the anatomic and functional organization of the fly and mouse olfactory systems have been mapped to excruciating detail, affording knowledge of how odors are processed along the entirety of olfactory pathway. In both model organisms, the layout of the olfactory system is two layers deep and comprises of a compression layer and an expansion layer. Olfactory perception is initiated by the recognition of odorants by a large repertoire of receptors in the sensory epithelium (Buck & Axel, 1991). In fruit flies, individual olfactory receptor neurons (ORNs) express only one of 50 different olfactory receptors (ORs), and all neurons (10 on average) that express the same receptor converge with precision onto a unique set of 2-4 projection neurons (PNs) through a specialized structure known as an olfactory glomerulus (Vosshall, Wong, & Axel, 2000). This layout establishes a one-to-one mapping between ORs and PNs. Information is then conveyed to an expansion layer of 2,500 Kenyon Cells (KCs) through sparse and random connectivity to support a high dimensional representation of odor information before it is classified by 20 read-out neurons, the mushroom body output neurons (MBONs). Experiments reveal that synaptic plasticity at the KC-MBON synapse is necessary and causal in odor learning.

The only major differences between the circuits of mice and flies appear to be numerical. Whereas the fly olfactory system consists of 50 ORs, 50 glomeruli, and 2500 KCs, the mouse olfactory system consists of 1500 ORs, 1500 glomeruli, and 1 million piriform neurons. The fact that evolution has evolved to hardwire the same architecture in flies, mice, and multiple other organisms suggests that such an architecture is optimal for the general task of odor sensing.

Although we have a detailed anatomy of the olfactory system in both flies and mice, it is unclear why certain features are optimal for odor sensing. In particular, 1) why does every ORN express a single OR, 2) why is information preserved through a one-to-one mapping between ORs and PNs, and 3) why is connectivity onto the expansion layer sparse and random (Litwin-Kumar, Harris, Axel, Sompolinsky, & Abbott, 2017)? To study optimal circuit design, we use a goal-driven approach to train an artificial neural network to classify odors and then analyze the anatomical and functional structures that emerge after training. This approach has recently been used to study the functional profiles of the ventral stream in visual object processing (Yamins & DiCarlo, 2016). The simplicity of the fly olfactory circuit and the exhaustive knowledge that we have of its anatomy provides constraints that can be used to gain insight into evolutionary design.

## Artificial neural networks and biological systems converge onto the same solutions

We use a simple odor classification task that maps odors to classes (100 total, 4 shown, Figure 1a). To generate this dataset, we first generated 100 odor prototypes. Each prototype activates 50 olfactory receptors, and the activation of each receptor is sampled independently from a uniform distribution drawn between 0 and 1. For all odors, the ground-truth class is set to be its closest odor prototype as measured by Euclidean distance in the ORN space. The training set consists of 1 million odors, and the validation consists of 8192 odors, and each odor is sampled the same way as the odor prototypes. Such a task mimics the evolutionary drive for organisms to distinguish between dissimilar odors and to generalize between similar odors.

The networks connections are modified during training to classify odors according to this exact mapping (Figure 1b). We used standard training techniques (Yamins & DiCarlo, 2016) based on stochastic gradient descent. This form of training can be thought of as evolving a circuit architecture in silico. We modeled the olfactory system as a layered feed-forward network with each layer corresponding to 500 ORNs, 50 PNs, 2500 KCs, and 100 class neurons, in this order. Connections between each layer represent synaptic strengths, and the activities of neurons in our network represent firing rates. The 500 ORNs are subdivided into 10 ORN duplicates that ex-

press the same OR for all 50 unique ORs. MBONs, the readout neurons of the olfactory system, are simplified into class neurons, and each outcome is represented by the activation of a single class neuron. For simplicity, this architecture omits several biological structures, including interneurons, a realistic readout, and an additional pathway.

Connections between neurons in each layer have no initial structure. After training, the network performs odor classification with 83% accuracy. We analyzed the network structure and observed that connections between ORNs and PNs appear to resemble the convergence of ORNs expressing the same OR onto glomeruli in the olfactory system. All ORNs that express the same OR project onto a unique PN, and PNs sample from a single type of ORN (Figure 1c). We quantify the extent that PNs pool from a single type of ORNs using a simple but stringent metric called Glomeruli Score (GloScore). A maximal GloScore of 1 means that PNs sample exclusively from ORNs expressing the same OR, whereas a score of 0 means that PNs sample from multiple ORs with the same connection weight. During training, the GloScore of the model ORN-PN connectivity quickly approaches 1.

Every KC is initially connected to all 50 PNs. While glomeruli are forming, we observe that these connections sparsen (Figure 1d). We measured the KC input degree (the number of strong PN connections for each KC neuron) after training and observed an average input degree of 7. This result is striking as it matches the input degree derived from exhaustive anatomical tracing studies (Caron, Ruta, Abbott, & Axel, 2013) (Figure 1e,f).

Theories suggest that random connectivity support high-dimensional representations that enhance the ability of downstream readout neurons to learn associations, much as in theories of cerebellar cortex (Babadi & Sompolinsky, 2014; Litwin-Kumar et al., 2017). We also observe that KCs evolved to randomly sample from PNs after training. We thus calculated the average correlation between the input connections of every pair of KCs, and found that correlations quickly decrease to approach that of randomly shuffled connectivity during training. A more stringent analysis revealed that KCs sample uniformly from all PNs and KCs in our network are not connected to any preferential pairs of PNs, similar to what has observed in the wiring diagram of fruit flies (Caron et al., 2013).

To ensure the robustness of these results, we performed an extensive hyperparameters sweep, exploring the impact of learning rate, the number of KCs, dropout rate, batch normalization, and input noise. The quantitative results are robust to all but one hyperparameter. Decreasing the learning rate eventually leads to a non-separation of weak and strong weights. Therefore, we used the highest learning rate that allows for classification accuracy to exceed 50% (chance is 1%). Moreover, these results were also robust to the addition of biologically realistic network motifs, such as normalization in the PN and KC layers.

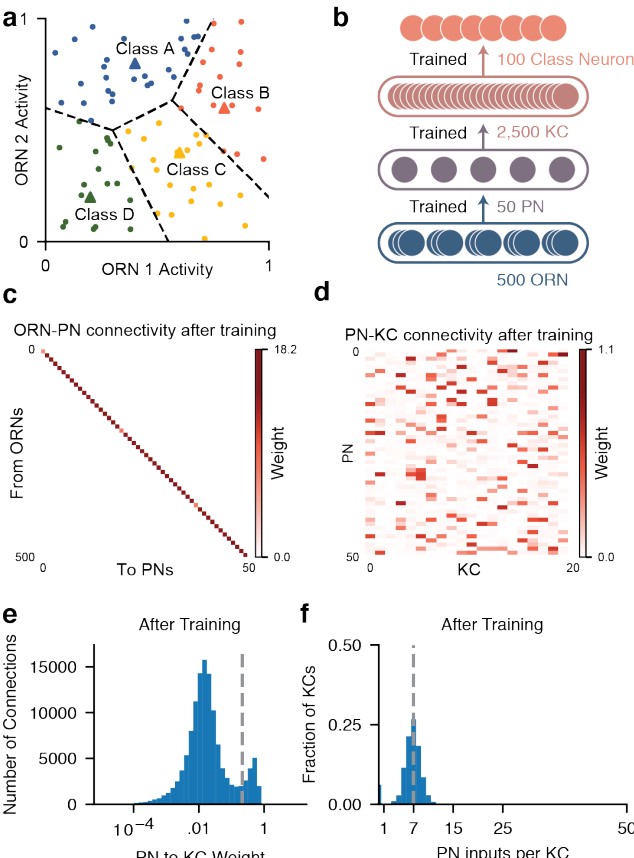

Figure 1: **Artificial neural network evolves the connectivity of the fly olfactory system.** (a) Schematic of dataset. In the space of ORN activity (50 dimensions, 2 dimensions shown), each test odor (a million in total, 100 shown in dots) is classified to the nearest class odor (100 in total, 4 shown in triangles). (b) Architecture of the artificial neural network. Each OR is expressed in 10 ORN duplicates. All connection weights are trained. (c) ORN-PN connectivity after training. ORNs are sorted by their receptor types. PNs are sorted by the strongest projecting OR. (d) The model PN-KC connectivity after training, only showing 20 KCs. (e) Distribution of PN-KC connection weights after training. (f) The distribution of KC input degrees after training.

## Emergence of uni-glomerular PNs that sample from unique ORs

In a network with exclusively excitatory connections, glomeruli emerge (Figure 1c). In a network with both excitatory and inhibitory connections, PNs mix from multiple ORs (Figure 2a). Irrespective of mixing, accuracy is maintained (Figure 2b). Moreover, the average Pearson's correlation between the activities of different PNs with and without mixing were close to zero, suggesting that connections to PNs evolved to preserve odor information. However, by minimizing correlation the network loses out on an opportunity to increase the dimensionality of its odor representation. We hypothesize that a network must preserve information if information can be expanded downstream. Conversely, a network with far less than 2500 KCs cannot expand and should bias the PN layer to mix ORs and increase dimensionality. We thus trained networks with variable numbers of KCs while keeping the numbers of ORNs and PNs fixed at 500 and 50, respectively. Indeed, as we decreased the number of KCs, GloScore decreases as PNs begin to sample from multiple ORs (Figure 2c). We further note that there is only a marginal benefit in performance with more than 2500 KCs (Figure 2d), which is the number of KCs within each hemisphere of the mushroom body.

We further predict that PNs will mix if given the resources to do so. We varied the number of PNs while keeping the numbers of ORNs and KCs fixed at 500 and 2500, respectively. Indeed, when the number of PNs exceeds the number of unique OR types, we observe that excess PNs receive mixed OR input (Figure 2e). Moreover, classification accuracy saturates, implying that having more than 50 PNs does not aid task performance (Figure 2f). When there are less than 50 PNs, information flow is bottlenecked, and mixing occurs to ensure that all ORs are represented (Figure 2e). Together, these results argue that the glomeruli representation is only optimal when expansion occurs downstream. We further added input noise sampled from a normal distribution onto ORNs, and found that the formation of glomeruli is minimally dependent on input noise.

## Sparse, random connectivity leads to maximum robustness

When trained on the standard classification task, the input degree of the expansion layer (KCs) settles at $K \approx 7$ (Figure 1f). To predict the connectivity of arbitrarily-sized olfactory systems, we trained networks with various numbers of ORs, ranging from 50 to 400. For each network, we quantified the input degree of the expansion layer after training (Figure 3a, plus signs). The results can be well fitted by a power law function, $\widetilde{K} \sim N^{0.71}$, where $\widetilde{K}$ is the optimal expansion layer input degree for a system of N unique ORs. This prediction is consistent with experimental estimates from anatomical studies in mouse, $K \approx 40 - 100$, $N \approx 1,000$ (Davison & Ehlers, 2011; Miyamichi et al., 2013) and fruit fly $K \approx 7$, $N \approx 50$ (Caron et al., 2013), (Figure 3a, x signs).

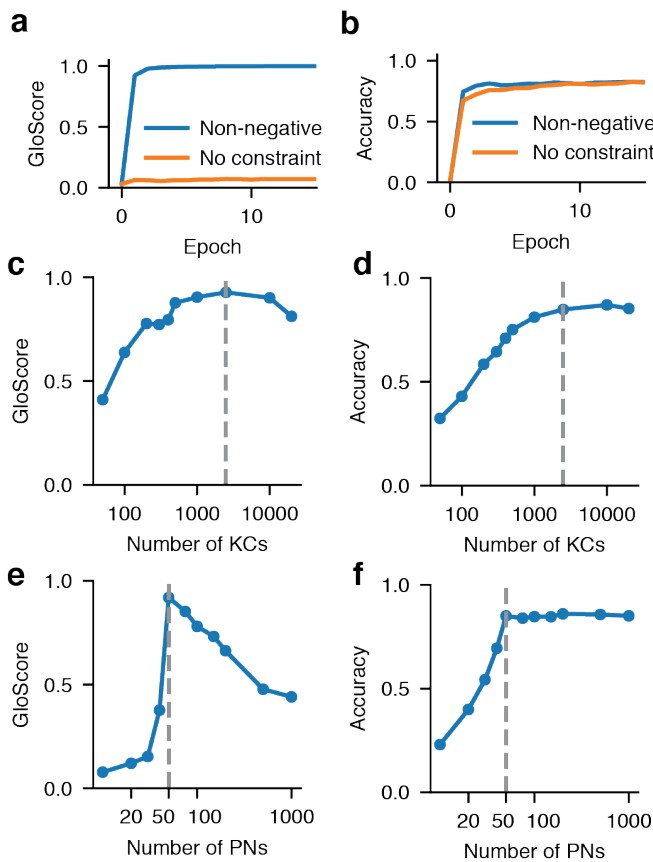

Figure 2: **Uni-glomerular connectivity emerges under biologically realistic conditions** (a,b) The GloScore (a) and accuracy (b) for networks with (blue) or without (orange) inhibitory ORN-PN connections as a function of training epochs. (c,d) The GloScore (c), and accuracy (d) after training for networks with different numbers of KCs. (e,f) The GloScore (e), and accuracy (f) during training with different numbers of PNs.

Next, we sought to understand what gives rise to the particular power-law of $\widetilde{K} \sim N^{0.71}$. The most obvious hypothesis is that these input degrees lead to optimal classification performance. However, we noticed that, for any $N$, a network with $K = 1$ can always be manually constructed to perform the standard classification task with 100% accuracy. Moreover, when N=50, networks fixed to have a wide range of $K$ values, from $K = 1$ to $K = 30$, performed the classification task with similarly high accuracy and low loss after training. Therefore, optimal classification performance cannot explain the development of $\widetilde{K} \sim N^{0.71}$. Neural representations with high dimensionality facilitates learning based on simple Hebbian rules (Litwin-Kumar et al., 2017), but optimal $\widetilde{K}$ estimates based on highest dimensionality predicts a power law with exponent around 0.3 (Figure 3a, gray), and more specifically, $\widetilde{K} \approx 8$ for $N = 1,000$, which is far lower than expected.

We hypothesized that an input degree of $\tilde{K} \sim N^{0.71}$ did not emerge solely to minimize classification loss. Instead, this power-law maximizes robustness to perturbations in connection weights. This hypothesis is inspired by findings that stochastic gradient descent, due to its stochastic nature, not only maximizes classification performance but also finds flat minima where the loss changes more gradually when connection weights are perturbed (Keskar, Mudigere, Nocedal, Smelyanskiy, & Tang, 2016). This robustness can be interpreted both as robustness against variability in synaptic transmission and also against variability between individuals of the same species. To quantify robustness, we measured the average angle between odor representations in the expansion layer before ($y$) and after perturbation ($y + \Delta$) to connection weights (Figure 3b). The perturbation angle is proportional to the amount of mis-classification that happens as a result of weight perturbation, so a smaller angle corresponds to a more robust network. When N=50 and K explicitly varied from 1 to 30, we found that the perturbation angle is indeed minimized at around $K = 7$ (Figure 3c), in agreement with our hypothesis. For a network with N unique ORs, we can numerically search for the optimal $\widetilde{K}$ that maximizes robustness (minimizes perturbation angle). We found that the optimal $\widetilde{K}$ derived from maximal robustness (Figure 3d, red line) matches closely with results from direct training of networks of various $N$ (Figure 3d, plus signs). While the

## Conclusion

We trained artificial neural networks using stochastic gradient descent to classify odors. We found that glomeruli emerged in PN layer, and sparse random connectivity emerges in the PN to KC connections. We then explored the sufficient conditions that enabled these features to emerge. We found that the formation of glomeruli did not depend on input noise but rather on the existence of an expansion layer downstream. In addition, we found that an expansion layer with a synaptic degree of 7 endows the olfactory system with robustness by allowing for large tolerances in synaptic efficacies without affecting task performance. Our work offers a framework to explain the

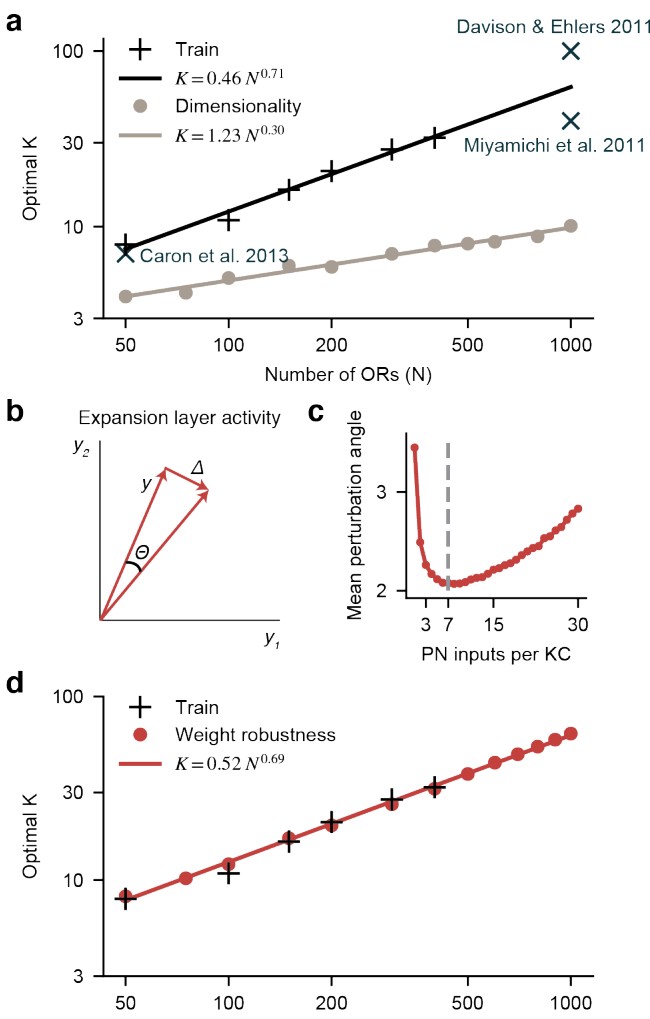

Figure 3: **Sparse PN-KC connectivity is robust to weight perturbation** (a) The optimal $K$ for networks with various number of ORs (N). Optimal $K$ predicted by direct training (plus signs), maximal dimensionality (gray), and experimental estimates (x signs). The lines are power-law fits. (b) Perturbation of connection weights from the compression to the expansion layer leads to perturbation $\Delta$ in the expansion layer representation $y$. (c) The average perturbation angle is minimized at $K = 7$ for fixed compression layer input degree K. (d) Optimal K predicted by maximal weight robustness (red) and direct training (plus signs). The line is a power-law fit of the red dots.

evolutionary convergence of olfactory circuits, and gives insight and logic into the anatomic and functional structure of the olfactory system.

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
