# OpenReview forum: "Evolving the Olfactory System"
_NeurIPS.cc/2019/Workshop/Neuro_AI — Real Neurons & Hidden Units @ NeurIPS 2019 Poster_

### Official Review · AnonReviewer2 · 2019-09-24
**Using SGD as an optimisation tool to find robust principles in a neuroscience model**

**Clarity:** 4

**Comment:**

This work is interesting and has nice conclusions, though its relevance to this exact workshop maybe a little off. It does not specify any sort of "idea cross pollination" from AI<->neuroscience. Still I think many in the attending audience will find it interesting.

**Category:**

Not applicable

**Clarity Comment:**

Nice clear paper.

**Evaluation:**

4: Very good

**Importance:**

3: Important

**Importance Comment:**

This work is important because it uses stochastic gradient descent (SGD) as an optimisation tool, rather than a 'learning model' as is so commonly done these days. The authors then argue that certain quantities in the model are robust to optimisation / optimal under certain constraints by converging to these parameters  values for a wide range of initial conditions.

**Intersection:**

2: Low

**Intersection Comment:**

This is really a theoretical neuroscience paper, I think.

**Rigor Comment:**

Though the space search is still numerical, the search methodology (i.e. optimisation) is adequate for the claims made.
An interesting step further would be to look at how 'sharp' posteriors over these parameters are (e.g. as in work like [Lueckmann, J. et al. Amortised inference for mechanistic models of neural dynamics; COSYNE2019])

**Technical Rigor:**

3: Convincing

---

### Official Review · AnonReviewer1 · 2019-09-25
**Understanding the olfactory circuit by perturbing it in silico**

**Clarity:** 4

**Comment:**

I think this paper is quite good, as provides a hypothesis on why the olfactory circuit has evolved its distinct architecture. I had some comments on what kind of data was used to train the model (real or synthetic), and also on if the same connectivity results persist if the model includes components such as divisive normalization and the APL.

**Category:**

AI->Neuro

**Clarity Comment:**

This paper is well-written, I have some small questions:
- Is the data used synthetic data, or is it based on a real dataset? (Hallem?)
- How was the data partitioned into training/validation/test sets?
- For the odor classification, what was the precision and recall across odors?
- "activities of different PNs are uncorrelated", how was this determined?
- At the end of a paragraph, "The formation of glomeruli is minimally dependent on input noise", what does this refer to?
- Fig. 3: Does this include changes to the number of PNs and KCs? Or just ORs?

**Evaluation:**

5: Excellent

**Importance:**

4: Very important

**Importance Comment:**

ANNs based on the olfactory circuit, which uses dimensionality compression followed by expansion, have already provided performance on nearest-neighbor lookup comparable with modern hashing algorithms. Continued research into understanding why the olfactory circuit has evolved its unique architecture could provide key insight into designing a new class of biological-inspired neural networks.

**Intersection:**

4: High

**Intersection Comment:**

This paper trains an artificial feed-forward neural network inspired by the architecture of the fly olfaction circuit. The authors perform experiments in silico and examine how the architecture of the model makes the network resistant to fluctuations in the weights.

**Rigor Comment:**

The findings in this paper appear strong, as they match experimental findings of PN-KC connectivity. I do have some questions about the model:
- Do you see the same connectivity results if you include an APL component in the model?
- Do you see the same results if you implement divisive normalization in the model? (Something like a softmax function between the ORN and PN)
- Does each odor class have the same number of odors, so it assumes that the fly encounters all odors evenly, or are some odor classes rarer than others?

**Technical Rigor:**

4: Very convincing

---

### Decision · Program_Chairs · 2019-10-02

Accept (Poster)

---

> ### Comment · ~Peter_Yiliu_Wang1 · 2019-10-28
> **Thank you reviewers**
>
> We thank the reviewers for their thoughtful and gracious comments and appreciate the time taken in reading our submission carefully.
>
> We have addressed these comments in the submission but will also respond in brief here:
>
> 1. Do you see the same connectivity results if you include an APL component in the model?
>
> Yes. An APL component was added in to either support feedforward and feedback inhibition, and did not affect our results.
>
> 2. Do you see the same results if you implement divisive normalization in the model? (Something like a softmax function between the ORN and PN)
>
> We implemented divisive normalization with learnable parameters, inspired by the Olsen and Wilson model, and this did not affect our results. Normalization was necessary for concentration-invariant classification, but was not needed for our task.
>
> 3. Does each odor class have the same number of odors, so it assumes that the fly encounters all odors evenly, or are some odor classes rarer than others?
>
> Odor activations for each olfactory receptor is sampled from a uniform distribution between 0 and 1. This was the same for training, validation, and test datasets. Therefore, it is not based on the Hallem/Carlson dataset, but completely synthetic. We sought to provide a general reason for why compression and expansion is optimal by using random data.
>
> 4. "activities of different PNs are uncorrelated", how was this determined?
>
> By calculating the the average Pearson's correlation coefficient between all pairs of glomeruli to odors.
>
> 5. At the end of a paragraph, "The formation of glomeruli is minimally dependent on input noise", what does this refer to?
>
> We tested the effect of adding Gaussian noise onto the activities of ORNs, and found that it does not impact the formation of glomeruli.
>
> 6. Fig. 3: Does this include changes to the number of PNs and KCs? Or just ORs?
>
> PNs were matched to the number of unique ORNs, but KCs were fixed to be 2500. An extensive hyper-parameter search revealed that the KC input degree (K) did not depend strongly on the number of KCs. For example, in the case of 50 ORNs, K = 7 when the number of KCs is 2500, and K=9 when the number of KCs is 10000.